# Virtual Reality in Space Technology Education

**Ghada Atta [1], Amal Abdelsattar [2],\*, Dalia Elfiky [1], Mohamed Zahran [1], Mohamed Farag [1] and Salwa O. Slim [3]**

[1] National Authority for Remote Sensing and Space Science, Alf Maskan, Cairo P.O. Box 1564, Egypt
[2] Department of Architecture, Prince Sultan University, Riyadh 66833, Saudi Arabia
[3] Faculty of Computers and Artificial Intelligence, Helwan University, Helwan 11795, Egypt
\* Correspondence: aabdelsattar@psu.edu.sa

**Abstract:** The simplification of space science and technology for students K–12 is a challenging task for educators. Virtual reality and augmented reality are educational techniques that introduce the concept of educational games. Moreover, those techniques have a stunning effect on students. This work presents the utilization of virtual reality models to teach students about the satellite types, satellite subsystems, the satellite assembly and integration process, watching the rocket launch carrying the satellite and observing the satellite in its orbit in virtual space laboratories. A 10-min mission in virtual laboratories will effectively improve the learning outcomes. In addition to the VR feature, a set of activities and short movies are considered to be beneficial for use by students to enrich the teaching results. Finally, the VR model results confirmed that the students' knowledge about the space technology cycle is boosted.

**Keywords:** virtual reality; space technology; education; game

## 1. Introduction

A virtual reality (VR) environment can be described as immersive multimedia or computer-simulated reality that allows users to interact with places in the real world or an imagined world in which they are physically present [1]. VR is a computer technology that provides a three-dimensional environment that naturally surrounds the user and responds to their actions.

VR is mainly a simulation that generates visual effects by using head-mounted display (HMD) systems. HDMs feature a wide viewing angle, head and hand movement tracking and objects that interact using controllers, so that a person can experience the virtual world on their head or as part of a helmet [2]. Through 360-degree videos, learners can observe scenes from any perspective they like. By using 360-degree video cameras, they can explore virtual worlds or view actual real-world recordings [3]. VR headsets allow users to view 360° content virtually instead of in their real environment, making them feel like they were there. As a result, 360° panoramic content creates an immersive experience that is more like a live feed than a regular photograph or video. Due to the panorama contents being mapped on a spherical surface as a single layer, viewers can view the panorama by moving their heads without engaging in any interactions [4].

Virtual reality was initially marketed toward gaming, but various sectors and fields can benefit from VR applications, such as industrial simulation [5], medicine [6], sport [7], social interaction [8], tourism [9], music [10], driving simulation [11], media [12] and education [13].

VR has gained considerable popularity in education over the last few years [14]. Virtual platforms are typically used to emulate the classroom or laboratory environment in which the classroom takes place. Nevertheless, some of these simulations are usually used as a safe environment to test scenarios that would be too difficult or dangerous for people to operate in the real world [15]. The use of virtual reality technology in education has a wide range of applications. Kavanagh et al. analyzed a total of 99 papers utilizing educational VR

software. The two most prevalent fields were health and engineering. However, VR is also used in other applications such as geometry, astronomy, interior design and history [16].

VR has gained considerable popularity in education over the last few years [14]. Many instructors have begun to use VR to execute difficult-to-practice jobs because there are not enough resources available or because doing so carries inherent hazards and dangers that can occasionally have catastrophic results. The most significant advantage of VR is that it gives students a chance to practice these skills in a secure environment while simultaneously immersing them deeply enough for it to feel realistic and transferable to the real world and properly reflect nearly any situation [1]. For example, students can walk on Mars or travel through the body through their blood cells [17]. In addition, VR can increase student attention by keeping them engaged with the VR environment [18,19]. Teenagers sometimes struggle to pay attention in class, especially when the topics discussed seem irrelevant to them. Immersing students in a virtual world such as VR increases their interest in and engagement with their learning [19,20]. Students can also focus better on teaching materials when wearing VR headsets since they eliminate visual and auditory distractions. By using VR approaches, teachers will have a greater opportunity to interact with students one-on-one and have more useful and beneficial interactions with students [21].

In addition, VR offers students a constructivist learning opportunity, which allows them to create knowledge by engaging in meaningful experiences. These experiences enable students to explore solutions to authentic problems and collaborate with others. Additionally, VR allows students to visualize and manipulate objects easily, enabling them to grasp difficult concepts more easily [22]. Researchers have proved that high-quality VR can improve academic performance and students' soft skills [23]. It has been shown that virtual reality can improve students' perception of space and depth in the 12 to 15 age group [24]. VR's immersive components can also help learners develop empathy for the subjects they experience [25].

Additionally, VR gives the students a clearer visual comprehension and depiction of their tasks, which can help them complete them more accurately, independently and with greater interest [26]. Interactive virtual experiences provide learners with greater exposure time without real-life difficulties, whether computer-generated or built on 360-degree video material. VR lets students learn from their mistakes. It gives low-skill students more training and access to learning materials than classroom time alone [3]. Several studies and reports indicate that most students remembered what they experienced in virtual reality, concluding that this environment is more memorable than traditional teaching techniques [27,28].

VR has presented a new way to teach astronomy and space technologies [29–33]. Using VR technology, Welsh and Windmiller introduced the concepts of introductory astronomy. To improve the learning experience, they developed immersive activities for two challenging "3D" concepts: lunar phases and stellar parallax [29,30]. In several studies, other researchers used VR to simulate the solar system [31–33]. Yair et al. examine the moon and Mars through observations, interpretations and comparisons. Geological and atmospheric processes are recognized and discussed, as well as astronomy phenomena, and students discover that the same fundamental physical laws apply to all solar system objects [31]. Mintz et al. offered a new interactive 3D model of the solar system in VR [32]. In a virtual world, learners can experience the physical world. The created virtual world continues to function as the normal world while the viewer zooms in or out and changes his/her viewpoint. In addition to allowing learners to travel in space, such an education tool creates a unique user experience [33]. No studies were found regarding the use of VR in teaching satellite types, satellite subsystems and the satellite assembly and integration process.

This research applies virtual reality technology to space technology education by simulating a space lab to assemble and integrate a satellite subsystem (CubeSat). The virtual space lab is designed to help the users gain a basic understanding about the satellite's subsystems and components in a simplified and attractive way.

This research aims to define the main concepts of space segments, satellite subsystems and components, assembly and integration of satellites, launching the satellite, controlling the satellite from the ground station and the resulting remote sensing images from the satellite. All those concepts are illustrated in a simplified way that depends on using virtual reality technology as a modern teaching method based on dazzling and creating suspense for the player using a simulation of the space labs. In this work, gamification was used in learning to improve training outcomes. Gamification is a method for applying game mechanics, elements and principles to non-game contexts to increase user engagement. Therefore, the target users are learners or students.

## 2. Virtual Lab Experience

### 2.1. The Hypotheses

In this work, we investigate how virtual reality (VR) can be used in space technology education, lecture and lab classes to provide authentic, student-centered research lab experiences to teach hard-to-understand 3-dimensional topics. By placing students in the space laboratory and letting them assemble the satellite modules, it is hoped that they will fully grasp these ideas. In this article, we detail how we have carried this research forward by developing a virtual reality (VR) lab activity focused on satellite manufacturing.

### 2.2. Scope of Work

As shown in Figure 1, the simulation design includes two stages. The first stage is 3D story-telling animation (duration: 5 min), which introduces the problem for the players to solve with the story's main characters. The second stage is the VR educational experience, which presents the theme, the space laboratories, the satellite types and subsystems, the assembly and integration of the satellite, and finally, the launching process and observation of the satellite.

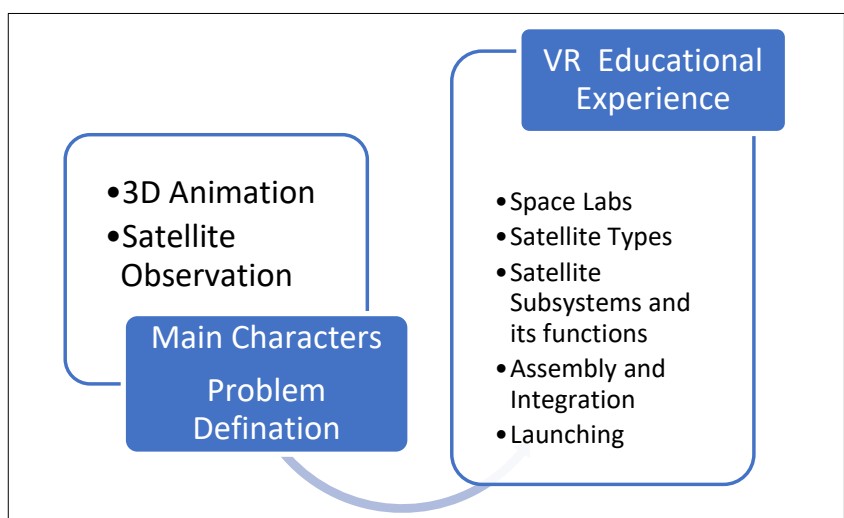

**Figure 1.** The overview of the VR simulation design.

### 2.3. Simulation Phases

As presented in Figure 2, there are four phases in the simulation. The first phase is the 3D animation video story, which defines the main characters and their problems. The second phase is the tutorial demonstration using VR glasses and hand controls to display the lab area. The third phase introduces satellite types, functions and subsystems and then defines the group mission that should be accomplished.



**Figure 2.** The simulation phases.

### 2.3.1. 3D Animation Video Story

This phase starts with a story-telling 3D animation video informing the students that the 3D alien characters are encountering technical problems with their spaceship's spectral camera, which will disable them from finding the resources that will save their planet. Therefore, they need help from the humans on Earth to have some earthly minerals to serve their planet. As shown in Figure 3, the NARSS professor, a professional in remote sensing satellites, suggests building a remote sensing satellite for the aliens with students' help.

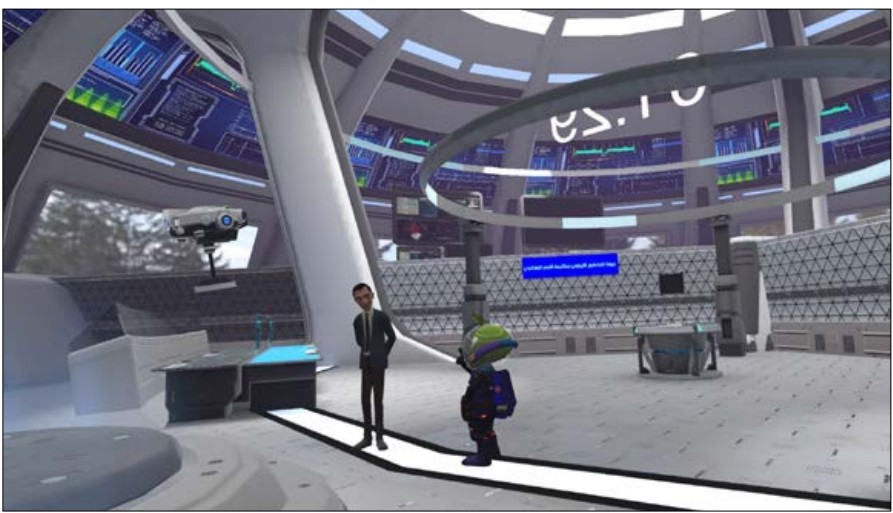

**Figure 3.** 3D animation video characters in the space lab.

The benefit of this phase is to define the role of the remote sensing satellite as a tool for mapping the Earth and finding minerals.

### 2.3.2. Tutorial Demonstration

The players will learn to use the VR hand controllers to select their avatars (color) in a specific order for selection and join the simulation. The space lab scene will be displayed directly after the tutorial demonstration.

This phase's purpose is to familiarize the students with new educational tools.

### 2.3.3. Satellite Introductory

An introduction to the satellite's types, functions and subsystems is presented in this stage. The NARSS professor illustrates to the students the roles of the remote sensing satellite, the weather satellite, the communication satellite and the deep space exploration satellite. Those satellite types are presented in an interactive hologram with their names. There will also be a voiceover (VO), as shown in Figure 4. The satellite segments, bus and payload are also presented in the hologram. The learning outcomes from this stage are learning the satellite types, each satellite's role and the function of each satellite module and its main parts.

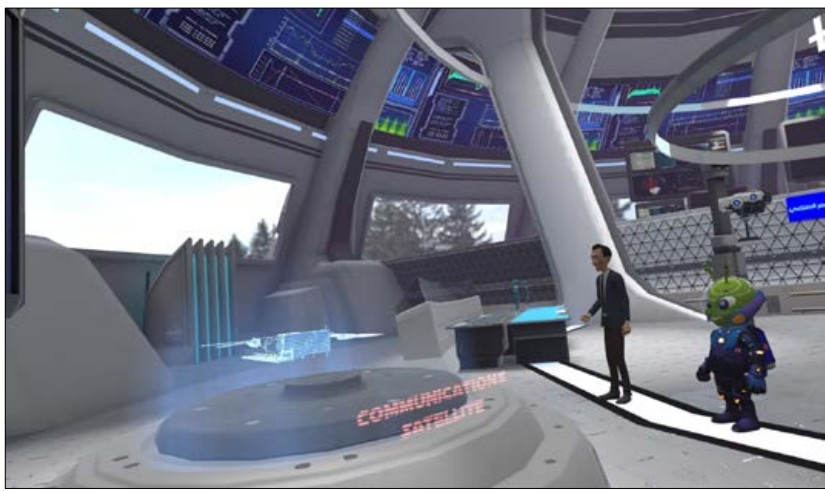

**Figure 4.** The hologram shows different types of satellites.

### 2.3.4. Students' Mission

The professor describes the work plan for the mission: assemble the satellite subsystems, integrate all subsystems to build the satellite and finally launch the satellite with the rocket and control it in its orbit.

Each student avatar is moving to the working bench to connect their system. The horizontal hologram screen appears to give them the main information about their satellite subsystem function and its parts. There are four main satellite subsystems: power subsystems, communication subsystems, onboard computer subsystems and payload subsystems. As shown in Figure 5.

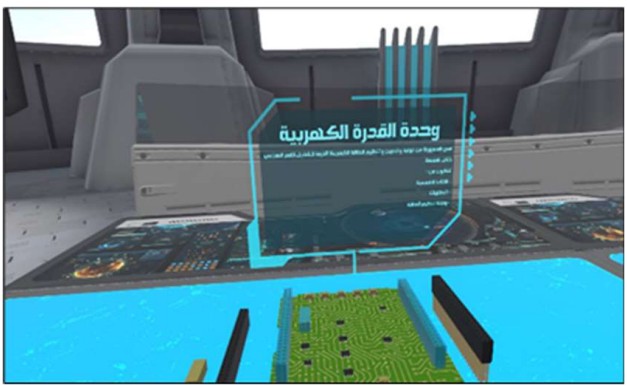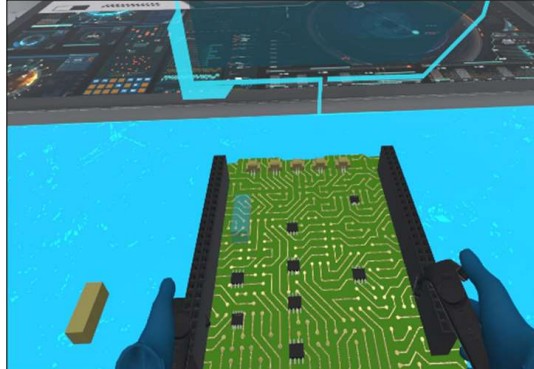

**Figure 5.** Assembly of satellite subsystems.

The student must fix each part in its position in the subsystem module to finish this level and upgrade to the next level.

The avatars are moving to the integration lab, where a 3D diagram of the remote sensing satellite parts will be displayed in this lab. Players can view and rotate the diagram with their hands. Once the avatars are near the integration bench, they will start integrating the satellite parts in a specific order. The players must try one after the other to find the correct part. If a player attempts to assemble the wrong part, it will not be inserted, and a red light will be displayed to instruct them to try another part. A green light will be displayed when the player puts the right parts together, permitting them to proceed. After the players complete their assembled parts, they will turn to gather at a central assembly point (a worktop) to put all the satellite parts together. A flash screen with the heading "Go into the control room" for each player is displayed in Arabic, as shown in Figure 6.

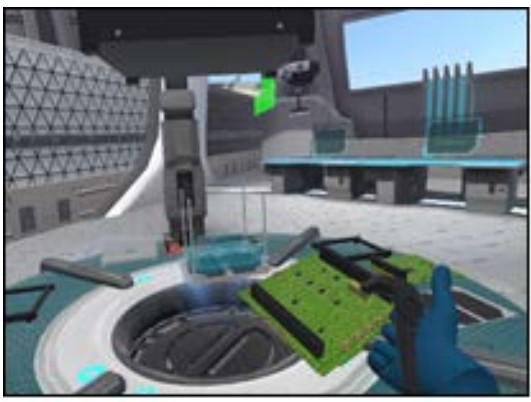
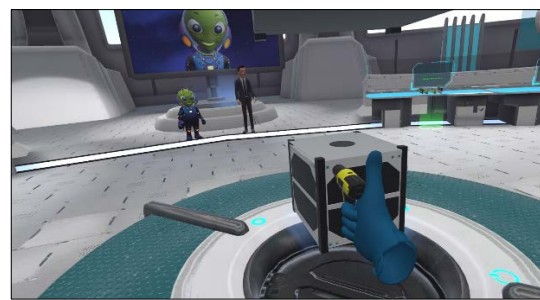

**Figure 6.** CubeSat after integration.

After the players complete integrating the satellite subsystems, the drone carries the satellite to the launch site, and the players go to the launch observation room (Figure 7).

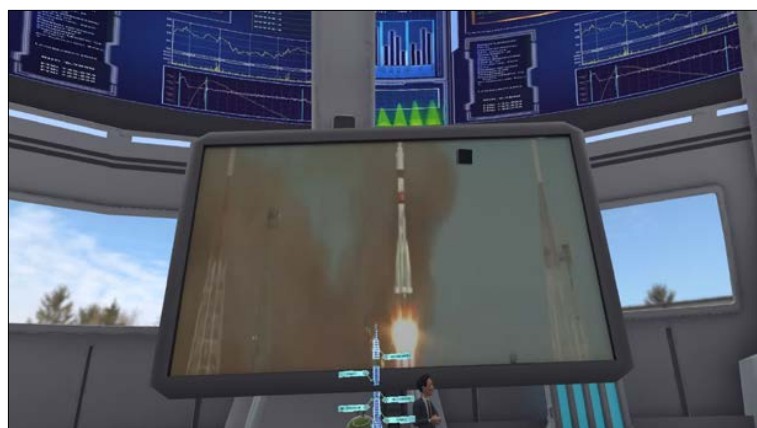

**Figure 7.** The launching observation room and hologram for the rocket.

The rocket parts are presented as a hologram, and the big screen shows the rocket carrying the satellite and ready for launching into outer space. Finally, the players have to enter the password for the launch.

After launching, the players have to move to the satellite control station to confirm that the satellite in their orbit is working very well and can send data and receive commands (Figure 8).

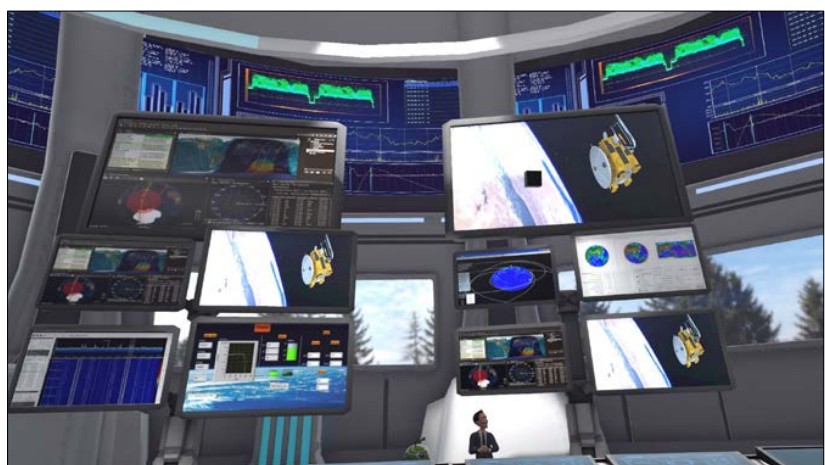

**Figure 8.** Satellite control station.

The learning outcomes of this stage are:

1. Understand the satellite manufacturing process.
2. Realize the main function of each satellite subsystem and its essential parts.
3. Recognize the integration procedure and the importance of fixing the module in a specific manner to accomplish the mission correctly.
4. Recognize the method of launching the satellite into the specific orbit using the rocket controlling the satellite in its orbit.

## 3. System Coding Components

### 3.1. Tools & Prerequisites

Virtual reality hardware usually consists of a head-mounted display set (HMD) and a computer device that is often separated from the HMD, such as the HTC Vive Pro VR system or integrated with the HMD, such as the Oculus Go VR system. In addition, controllers and audio devices are attached to extend the functionalities of VR sensors. Recently, smartphones have entered the VR market as an alternative to computers, but with a big performance gap. Virtual reality software consists of an SDK that highly depends on the target hardware. In our proposed system, we use SteamVR SDK (OpenVR), which contributes to making a generic SDK for most hardware such as open XR, but it still needs more enhancement. The other software part is the game engine, and most of the creative work is performed by it. The Unity Game Engine makes virtual reality development easier than before. So, there are two main components to begin the game, Unity editor version 2019.1 and the HTC Vive headset and controllers.

### 3.2. Modules

The game system consists of different modules. Figure 9 shows the relationship between those modules.

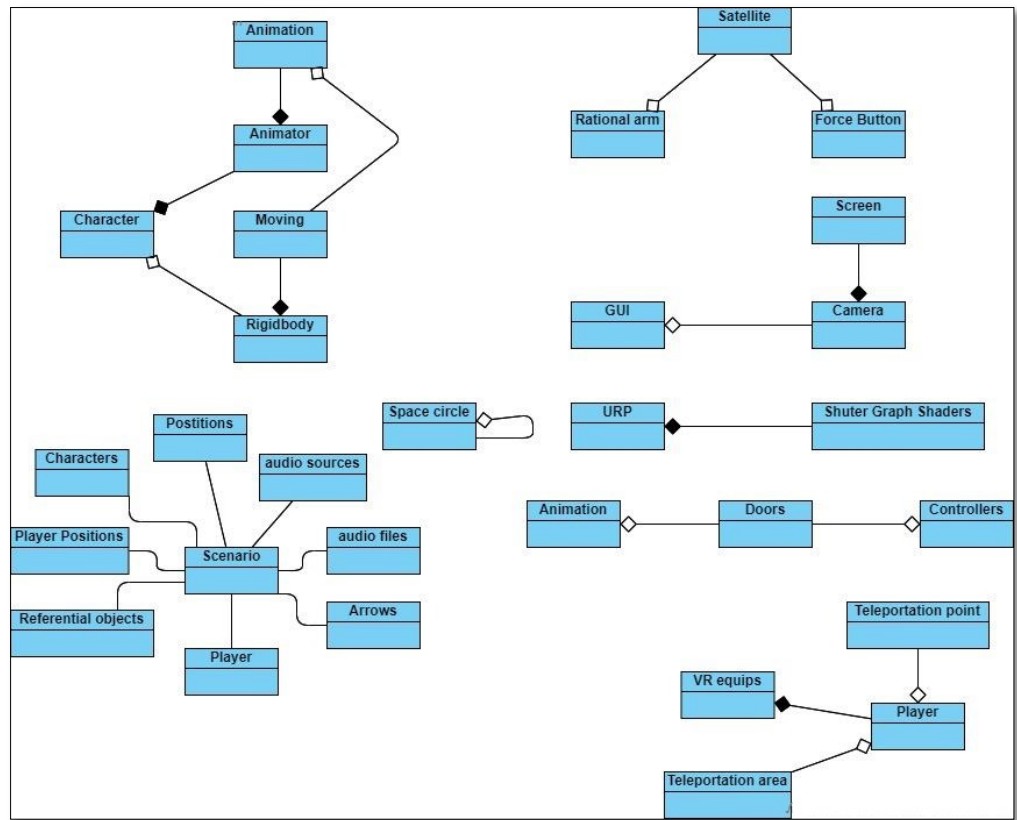

**Figure 9.** Layers and scenario modules.

Layers module: This game has two main layers separated from each other; the first layer is the control center, and the second layer is the space. The purpose of the two layers is to separate the cameras' outputs in the space rather than the cameras' outputs in the control center. Elements of space are satellites, rocks, geomagnetic fields, portals, explosions and fire effects. So, the cameras in the control center layer could not see those elements and vice versa in the space layer. The control center screens display all game characters and cameras' views of the whole game accurately and realistically.

Scenarios module: The control center contains three rooms, each with a scenario. The scenario is a mix of different components and events that will take place in a particular sequence to allow the player impact the game and enable the level to be progressive. Therefore, we created a template that helps us to handle the scenarios and the events perfectly in the sequence we want. The template is a script that contains several variables, functions, a start function and an update function.

Cameras Module: There are many cameras on the scene. Each camera records a specific view and has a particular GUI. Each camera's output is exported as a render texture, and the render texture is used in a material connected to a specific screen in the room. There are three types: the first type is the cameras that do not have a GUI. They return a view, and they are used in the cameras that are in the control center. The second type is the cameras that have a static GUI, which is a GUI that is not attached to a script, so it just has some text with some colors, and it is used in the terminal camera. The third type has a GUI attached to the script to make it updatable, such as the screen that displays the speed, temperature and ping of the satellite and the screen that displays the satellite status.

Satellite module: The satellite has a system for movement, health, data, damage, and destruction. This system manages the satellite's behavior and allows the player to control it easily. Two main functions operate the motion of the satellite. The first function gets the distance between the rotation hand's center in the control center and its new position, then converts this distance with a threshold to a rotation used to rotate the satellite. The second function is a Boolean function that determines if a button is pressed or not to add force in the same direction as the satellite. It will not work if the satellite is destroyed. The satellite's health starts at 100% when the level is started but decreases by 10% if space rocks are hit. Therefore, the satellite's health is a major variable in detecting the satellite's current temperature and knowing whether the satellite controls should work correctly. Data of the satellite are represented as its health, temperature, velocity, ping, position, and rotation. Damaging the satellite will start a fire effect and decreases its health. When the health reaches 0, the satellite will be destroyed, and its parts will be devastated in space by instantiating new parts instead of the old ones. This is because they have independent, rigid bodies that will take the explosive force of the destruction and fly in different directions.

Player module: The player as a VR object has five main parts (VR headset, VR controllers, head collider, hands collider and body collider). The player can visualize the 3D virtual world using the VR headset. VR controllers are used to teleporting from one position to another and can pause the game with it. The head collider prevents the player's head from moving through walls. Hand colliders allow the player to interact with intractable objects like a rotational arm or force button to control the satellite's movement. Finally, the body collider is used to open the control center doors when the player is in their range.

Character Animation module: Every character has a unique animator that matches their rigging and follows the same animator technique. There are transitions from the "Any State" state to any other state. We control the animation sequence by changing the AnimationState value parameter to the state's entry value. Still, for every character, the entries are different. For example, the bald entry for idle animation may be when AnimationState equals 1. Still, the entry for idle animation for the black character is when the AnimationState equals 5.

Character Movement module: This system has four main components (rigid body, movement Boolean, movement speed and destination position). When the movement Boolean is true, the rigid body will have a velocity equal to the movement speed variable

directed to the destination position. The character will move with that speed from its position to its destination. When the character is near the destination by some distance, the velocity will turn to vector zero to make the character stop and then set the movement Boolean to false.

Teleportation and Door module: We put teleportation points and areas along the control center. There are two invisible triggers on the front and back of every door. When the player's body collider enters these triggers, the door will open, and if the player leaves these triggers, the door will close. In addition, there are teleportation points over these triggers to make it easier for the player to detect their position. The player can teleport to any teleportation point or any teleportation area by pointing to it with the VR controller and pressing the teleport button.

## 4. Results

We implemented a simulation experiment for a space lab using virtual reality technology in the NARSS VR Academy to measure the effectiveness of the experience on learners through a field visit.

The practical experience of this research was applied to two groups of children in Nile International Schools for the fifth grade of primary school, and their ages ranged from 10 to 12 years. The first group received the concepts of space sciences through a theoretical lecture for 80 students in the school; the second group received the concepts of space science by visiting the virtual reality lab at NARSS for 100 students. To measure the effectiveness of receiving information for children using traditional methods of education compared to the use of modern technology in education, such as virtual reality.

### 4.1. The Control Group

The researchers made a field visit to the Nile International School to give a theoretical lecture in the form of a presentation to clarify the concepts of space science for 60 min. First, students were tested with five simplified questions about satellites before the lecture (Figure 10) to measure their knowledge. Then, the same test was done on the students after the lecture to measure the extent to which they had received information after the theoretical lecture. Students' answers before and after the experiment were corrected for five questions, each with two marks. The improvement in student answers after the theoretical lecture is shown in Figure 11.

**Pre-experiment Questionnaire**

**Choose the correct answer**

1. Communication satellites are used in
   a. Phones and television
   b. Earth observation
   c. Computers
2. The CubeSat consists of
   a. Number of 6 plates
   b. Number of 7 plates
   c. Number of 4 plates
3. The electrical power unit in the CubeSat satellite is responsible for
   a. Contact with the satellite
   b. Connecting the electrical power to the satellite
   c. Transferring the data from the satellite to the earth station
4. The space payload unit in the Cubesat satellite is responsible for
   a. Run the satellite
   b. Satellite launch
   c. Photographing the Earth
5. Remote sensing satellites provide
   a. Information about outer space
   b. Data information on weather
   c. Satellite image

**Figure 10.** Questionnaire before the experiment.

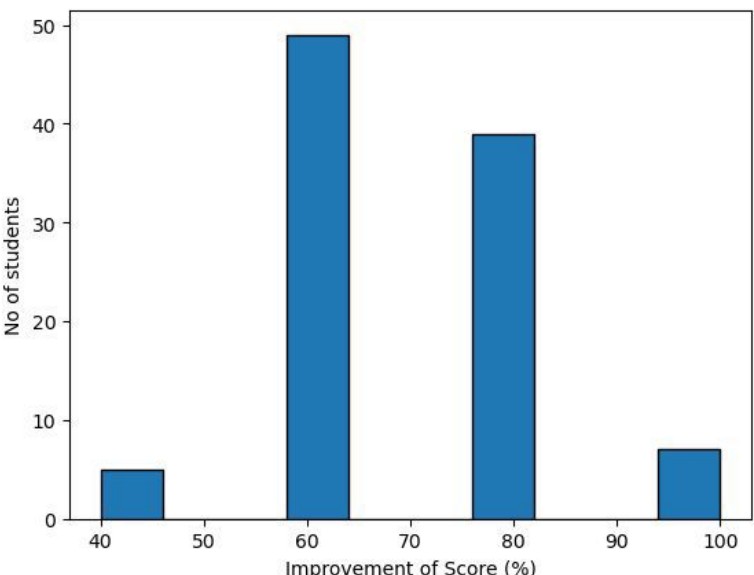

**Figure 11.** The histogram for the score improvement achieved by the lecture for the students.

The histogram shows that about five students enhanced their space knowledge by 50% and about 95 students by 60–100%. However, by analyzing the previous histogram, the students did not answer the questions because they did not know the answers or only knew the answers to one or two questions. After the experience, one or two or three questions at most were answered, and some students did not answer because they were not interested in the theoretical lecture. This indicates that the knowledge of space sciences received by students through theoretical lectures is insufficient.

### 4.2. Field Visit

A field visit was offered to 100 students from the Egyptian Nile schools in the fifth grade of primary school (Figure 12), from the age of 10 to 12 years, and the tour was in two phases:

- Watching a 3D cartoon video of space technologies for 5 min.
- Virtual reality simulation experience lasting 10 min.

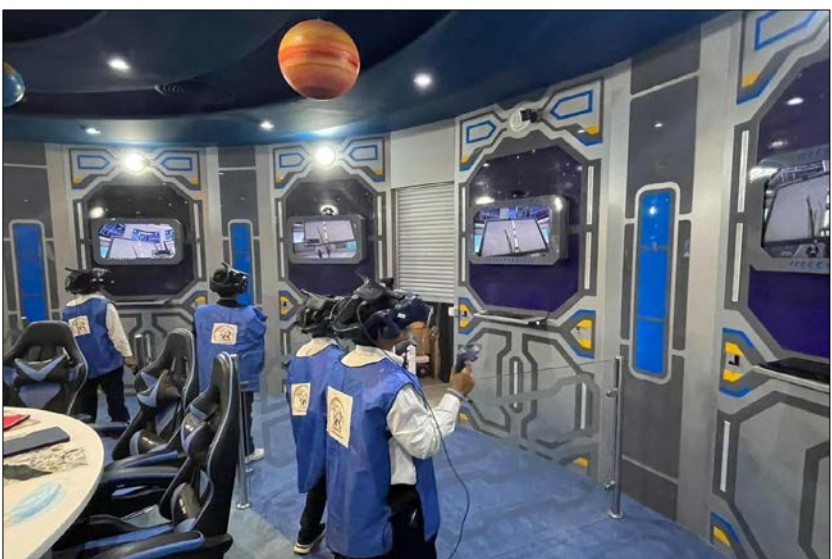

**Figure 12.** The students from the Egyptian Nile schools were in the fifth grade.

### 4.3. Measuring the Effectiveness of the Experience

The measure of the effectiveness of the students' benefit from the experience was carried out in two stages:

**Stage 1: before the experience**

Students were tested with five simplified questions about satellites before the experience to measure the extent of their knowledge (Figure 10).

**Stage 2: After the experience**

The students were asked to answer the same questions they solved before the experiment and to answer a questionnaire to measure their satisfaction with the experience (Figure 13).

**Post-experiment Questionnaire**

| After the Experiment | | Yes | Maybe | No |
|---|---|---|---|---|
| 1 | It was easy to understand the contents of VR | | | |
| 2 | The scientific content of space is simple and attractive | | | |
| 3 | Using VR glasses was easy | | | |
| 4 | I hope to try again | | | |
| 5 | I have gained information about satellites from my experience | | | |

**Choose the correct answer**

1. Communication satellites are used in
   a. Phones and television
   b. Earth observation
   c. Computers
2. The CubeSat consists of
   a. Number of 6 plates
   b. Number of 7 plates
   c. Number of 4 plates
3. The electrical power unit in the CubeSat satellite is responsible for
   a. Contact with the satellite
   b. Connecting the electrical power to the satellite
   c. Transferring the data from the satellite to the earth station
4. The space payload unit in the Cubesat satellite is responsible for
   a. Run the satellite
   b. Satellite launch
   c. Photographing the Earth
5. Remote sensing satellites provide
   a. Information about outer space
   b. Data information on weather
   c. Satellite image

**Figure 13.** Questionnaire after the experiment.

### 4.4. Student Answer Analytics

The students' answers before and after the experiment were corrected for five questions, each with two marks. Before the experiment, many students did not answer the questions because they did not have any space knowledge, while a few students answered one or two questions only as they had some information from the internet. After the experiment, at least three questions were answered, and many students answered all the questions (Figure 14). This indicates an increase in the knowledge of satellites among students after the experiment, which achieves the goal of its establishment.

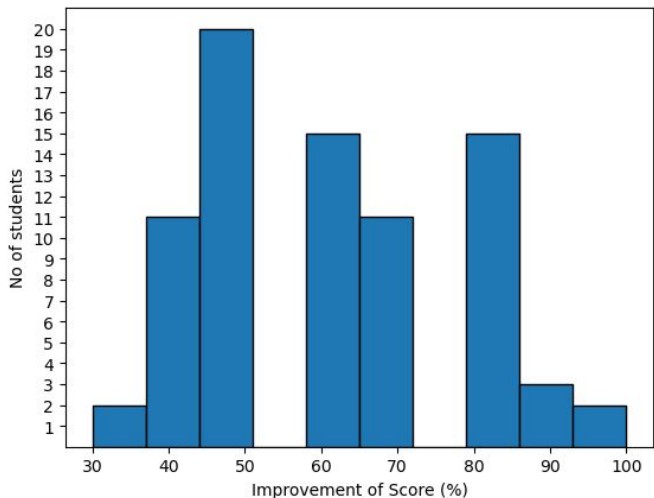

**Figure 14.** The histogram for the score improvement achieved by VR for the students.

The histogram shows that about 30 students enhanced their space knowledge by 50% and about 80 students by 60–100%.

A questionnaire was made for students to measure their satisfaction with the experiment. The students did not find it difficult to understand the information; it was simple and attractive. Furthermore, many of them did not find it challenging to use glasses (Table 1, Figure 15), and most of the students increased their knowledge about satellites and their components, which achieved the goal of the establishment.

**Table 1.** Analysis of students' satisfaction with the experiment.

| | After Experiment | | | |
|---|---|---|---|---|
| # | Virtual Reality | Yes | Maybe | No |
| 1 | It was easy to understand the contents of VR | 80 | 16 | 4 |
| 2 | The scientific content of space is simple and attractive | 75 | 18 | 7 |
| 3 | Using VR glasses was easy | 67 | 22 | 11 |
| 4 | I hope to try again | 85 | 15 | 0 |
| 5 | I have gained information about satellites from my experience | 90 | 7 | 3 |

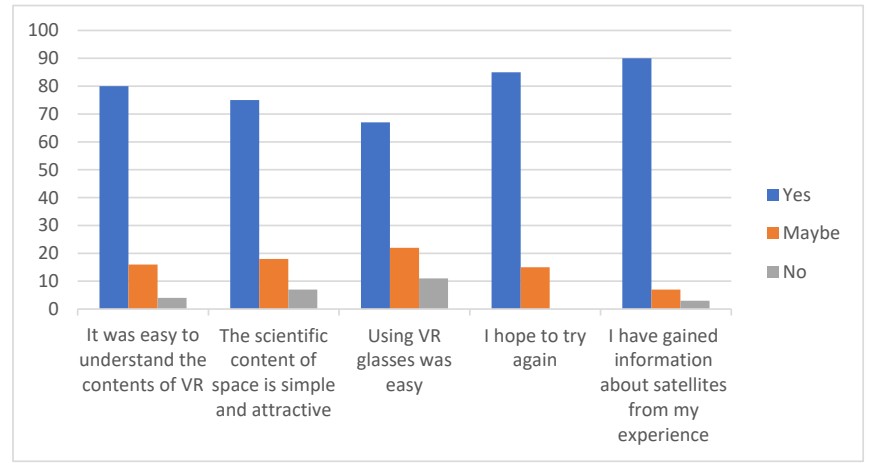

**Figure 15.** Graph of students' satisfaction with the experiment.

## 5. Discussion

VR is capable of revolutionizing the learning experience. It gives students access to experiences they otherwise would not have had [18]. The spread of VR technology was initially centered around games, but now VR is used in various sectors, such as education, medicine, health and engineering [27]. Furthermore, with the recent revolution in information and communications technology, teachers and students have access to many new teaching methods and tools, such as VR [34].

This study used VR to simplify space technology concepts for K–12 students. In addition, this study used VR to develop educational means that combine science and fantasy to motivate children, build their skills, and establish their capabilities of imaginative thinking.

Evaluating the VR trial was based on comparing the test scores of 100 students before and after the experiment. Moreover, students have to fill in a survey after the experiment. The trial showed that the VR gaming group performed better on the post-test than on the pre-test and even better than the control group. To evaluate VR in education, the rubrics should cover virtual representation, learning theory, user experience and human interactive design domains [17]. The survey for the VR trial was designed to cover most of these domains. Eighty percent of respondents said they had no trouble following up with the experiment since 75% reported that the information was "simple and interesting", 67% said they had no problem with the glasses and 90% said they gained knowledge about satellite from experience. Students enjoyed the VR experiment and were willing to repeat it as, according to the survey, 85% of the students were keen to repeat the experiment.

Converting the classroom into a dynamic learning environment has proven to be an effective and sustainable method of educating children. In addition, the immersive nature of VR increased student engagement and motivation through the use novel instructional formats and strategies. Furthermore, the stalemate and abstraction associated with many conceptual pieces of knowledge were broken, and the learning process was made understandable and fun by presenting scientific topics in a friendly manner. Still, getting VR widely adopted in classrooms and other educational settings is challenging. For VR to work well and provide worthwhile immersion and interactivity, it often requires sophisticated graphics capabilities that are not always feasible with regular computer hardware [1]. The use of virtual reality (VR) in virtual laboratories has emerged as a method to facilitate the learning and teaching process for a wide range of training activities [35].

In addition, VR needs full packages, including installation, maintenance, and technical support. Despite that, they come with a cost that not every teacher, instructor, or educational institution can afford [34]. Experts in the field of space were required to create scenarios to connect business owners, end users and designers. It also needs a group of programmers to implement the experiment and turn it into a virtual learning lab. VR is sometimes viewed as a game that is not taken seriously; it is enjoyable to play with but not a genuine learning tool. Students may have a winning mentality but may not completely engage their minds to learn new information and exercise critical thinking [1].

## 6. Conclusions

The educational field is one area where virtual reality can be used in novel ways. Incorporating VR into the classroom is a great approach to getting kids interested in and excited about learning about space technology. Simulating real-lab conditions as closely as possible promotes student learning by activating multiple perceptual channels [10–13].

This study presents the use of VR in teaching students about satellite types, satellite subsystems, the satellite assembly and integration process, watching the rocket launch carrying the satellite and observing the satellite in its orbit in virtual space laboratories. Due to its novelty, this approach to teaching kids still lacks the requisite groundwork and resources. Students were involved in the development and testing of a brand-new VR game. The trial conducted at the VR NARSS Academy confirmed the game's utility in teaching space technology. Based on the results acquired, this study's main conclusion is that VR positively affects learning among K–12 students in the space technology field.

Furthermore, when VR was compared with the traditional method of education, it was proven that VR improved the learning process.

**Author Contributions:** Conceptualization, G.A. and M.Z.; methodology, G.A.; software, S.O.S.; validation, G.A., D.E., M.F., A.A. and S.O.S.; formal analysis, S.O.S.; investigation, G.A. and M.F.; resources, M.Z. and A.A.; data curation, G.A.; writing—original draft preparation, G.A.; writing—review and editing, A.A.; visualization, M.F., D.E. and S.O.S.; supervision, G.A. and M.Z.; project administration, G.A.; funding acquisition, A.A. All authors have read and agreed to the published version of the manuscript.

**Funding:** This research is funded by Prince Sultan University, Riyadh, Saudi Arabia: 2022.

**Institutional Review Board Statement:** Not applicable.

**Informed Consent Statement:** Not applicable.

**Data Availability Statement:** The data presented in this study are available on request from the corresponding author.

**Acknowledgments:** The authors would like to acknowledge the support of Prince Sultan University for paying the Article Processing Charges (APC) of this publication. Moreover, the authors would like to acknowledge the National Authority for Remote Sensing and Space Sciences, Egypt, for the place, devices, equipment, assistance and support provided to them.

**Conflicts of Interest:** The authors declare no conflict of interest.

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
