# Peer review of "Virtual Reality in Space Technology Education"

_education, doi:10.3390/educsci12120890_

Round 1

Reviewer 1 Report

The authors present work developing a virtual reality application to teach students about satellite systems and launches. A short movie also accompanies the activity. Results from using the VR application with 100 fifth grade students from the Egyptian Nile schools is documented, along with some pre- and post-experience survey data to measure the effectiveness of the experience.

The premise of the paper has considerable merit. Although VR is being adopted in a variety of educational settings, relatively few studies have been published about its effectiveness in achieving desired learning outcomes -- especially in various areas of space education.  However, as submitted, the manuscript requires significant revision before I can recommend publication in Education Sciences. The content is largely descriptive of the VR app being used and does not provide results that can guide readers who wish to tailor VR experiences for their own applications. The hypothesis being tested is unclear and the actual study of students does not supply particularly informative results. It appears as though many students did not complete all questions, which raises concerns about the reliability of the data. It's also unclear how well the students retained the information they learned.  

The text is poorly referenced and does not cite relevant literature needed to set the context for the paper. Too much content is dedicated to describing the storyline of the VR activity and its coding, and not enough content explains what the authors set out to achieve and how well they were able to achieve those goals (especially in the introduction). 

This paper may merit publication if the authors take these general concerns into consideration. Most concerning is that the paper fails to explicitly define a central hypothesis to be tested, and includes too much technical material irrelevant to learning goals. Below are additional specific comments that should be considered in a new draft: 

1. Throughout the text "he" is used to reference participants. Instead use gender neutral references such as "he or she" "she/he" or simply "participants".

2. The manuscript should be read over closely for proper English. For example, many times "lunch" is inappropriately used instead of "launch".

3. The necessity of material presented in section 3 "System Coding Components" is unclear: the introductory paragraph of this section is general and vague, and the detailed description of modules does not contribute to enhancing interpretation of the results. Although it is clear that substantial work has gone into the creation of the VR application (including storyline, graphics, and coding), it seems that this section of the paper (which makes up a significant portion of the manuscript) is better suited for another journal.

4. The question "Use VR glasses" of the questionnaire is confusing. Did the authors intend to ask "VR glasses were difficult to use?" 

5. Lines 415-416. Can the authors elaborate on what challenges confront use of VR in educational settings?

6. Lines 425-426 "The trial showed that the VR-gaming group performed better on the post-test than the other two." What "other two"?  

Author Response

Thank you for the valuable comments. The author's response is in the attached file.

Reviewer 2 Report

GENERAL COMMENTS

The authors have some references to other VR-related research in their introduction, but some obvious ones are missing, and some that are included are not quite relevant. Find some that are published since 2016 (after the Oculus Rift CVcame out) related to space, satellites, and/or non-medical STEM. 

The virtual experience they have designed is very interesting! It is worth publishing something about this simulation.

The details in section 3 about the coding are not appropriate for this sort of article. Readers don’t need to see every variable in your Unity project. This feels like it was written to fill up space, especially in section 3.2 and Figure 10.

The research results based on the five simplified questions does not seem to be experimentally or statistically sound. The authors don’t appear to understand how to do this sort of educational research analysis and might want to either read other research on this or bring in a collaborator to help them.

The results in Figure 14 are confusing since the axes aren’t labeled and the scoring isn’t described.

The authors are not citing-and don’t seem to be aware of--any similar VR educational research at all after their introduction. The methods, results, discussions, and conclusions should all connect to other published research.

In conclusion, I do think a description of this VR project is worth publishing. And the English in this paper only needs a bit of fixing. But the authors seem to lack an awareness of how much detail is needed for educational research. They need much less detail about the inner workings of Unity and much more about the statistical validity of their results and how their findings compare to other similar research.

SPECIFIC COMMENTS

Nearly all users of virtual reality abbreviate it as VR not as V.R. when writing about it.

Citation 1 is spelled Shavinina, not Havinina. That article is about innovation in general but does not directly mention VR.

Read Steuer (1992) for a good psychology (rather than technology) based definition of VR beyond the “goggles and gloves” definition. That said, your own work should focus on other research that uses “goggles and gloves.”

I’m not sure what a “Virtual Round” is that is mentioned on line 37.

I couldn’t find any mention of citation 3 in Google or Google Scholar. What is it?

Citation 5 is about “desktop virtual reality” which is based on Steuer’s old psychology-based definition. It’s probably not appropriate to cite it here since modern researchers probably don’t consider that to be “virtual reality” since the mid-2010s.

Author Response

(The authors gave the same response as above.)

Reviewer 3 Report

Please see the review in the attached PDF. I was confused by the Conclusions that indicate there were (two?) comparison/control groups used to test the efficicacy of the VR experience, but I could not find out more about them in the manuscript.

Author Response

Thank you for the valuable comments. The authors' response is in the attached file.

Reviewer 4 Report

The authors investigated a novel VR method for space technology education. The proposed method shows may some merits in the field. The reviewer has a few comments that may be helpful to improve the paper:

1) Although some discussions are offered in the paper regarding the related VR work in the literature, the existing work on VR applications in space education is not well investigated in the paper. The authors shall add a comprehensive review and point out how the new VR method in this study could bring value to the field.

2) Section 2 seems to be packed with multiple topics. The reviewer only sees Section 2.1 while there is no Section 2.2, 2.3... Section 2.1.3 shall be revisited and improved as the reviewer still could not follow them after reading them twice.

3) Is Section 3 about the description of VR tutorial development or the description of what the students shall see in VR? It is very challenging to read this section. What is the VR engine adopted in this study? How the VR environment is modeled? Roughly, how many hours are needed for developing a VR in this study?

4) In terms of the evaluation, there are a few existing works on how to design the assessment instrument on evaluating students' virtual experience (e.g., rubric for eduVR, see the paper below). Could the authors add some discussions in this regard to the paper?

Fegely, A., & Cherner, T. S. (2021). A comprehensive rubric for evaluating EduVR. Journal of Information Technology Education. Research20, 137.

5) A critical concern from the reviewer is that there is no control group in the assessment. What is the performance of the student if he/she does not attend the VR section but studies the material using traditional methods (e.g., reading a book, face-to-face lecture, etc.). This would be a major weakness of the study as it is hard to prove that VR is more effective than traditional instruction approaches. 

6) The reference list is short while the reviewer likes to see a more deep literature review. One aspect that could be added to the paper is to review VRs that are based on 360-degree videos of real-world experiences. Studies in this domain have brought increased interest in the field. Some of the work is in the field of medical education, chemistry lab instruction, and additive manufacturing education. Adding these discussions would strengthen the quality of the paper.

Author Response

(The authors gave the same response as above.)

Round 2

Reviewer 1 Report

The updated manuscript has been improved substantially. The paper now clearly defines the hypothesis being tested, better provides introductory material and associated references, and overall now has a structure that is easier to follow. 

The introduction now provides additional context for use of VR in education. However, some of this introduction can be omitted or better clarified. For example the meaning of this sentence is unclear: "Despite the high costs associated with the initial setup, these experiences can be financed by investments."

The hypothesis being tested is now made more clear to the reader. However, the first and second sentences of section 2.1 seem redundant. Perhaps the "Produce an interactive edutainment experience ..." can be omitted and the section begin "In this work we investigate how Virtual Reality (VR) can be used in space technology education ..."

Line 172, "... the drone cries the" ==> "... the drone carries the ..."

Author Response

Thank you so much for your valuable comments.

Kindly find attached the authors' responses to your comments.

Reviewer 2 Report

This is an improvement on the original version. However, some large chunks of content are not appropriate for publishing in an education journal, and there are some pieces missing that should be there.

Nearly all of Section 3.2 is not the sort of thing one publishes. You might publish lines 234 to 248 (with specifics about the satellite module). You could also move section 3.1 Tools and Prerequisites to just after 2.1 but before 2.1.1. However, most of 3.2 includes Unity details that aren't appropriate for an education article.

This paper also lacks basic steps typically included in statistically sound education research, specifically in validating the survey questions, presenting the results, and calculating uncertainties. I'd suggest reading Slater's article on "The Development And Validation Of The Test Of Astronomy STandards (TOAST)" to get a feel for what is appropriate.

The plots created are not done in a useful format. Consider using histograms or perhaps x-y scatter plots showing pre- vs. post-VR scores.

Author Response

(The authors gave the same response as above.)

Reviewer 3 Report

I appreciate all the work that the authors put in to this version. It has resulted in a much better paper and will no doubt be a useful resource for other educators who embark on the mission of using VR for space edcuation.

I only have a complaint about Figure 10 - it is confusing to the eye and independent data points should not be connected with straight lines. Instead, it and Figure 14 should use the same style of presentation.

Author Response

(The authors gave the same response as above.)

Reviewer 4 Report

The authors have addressed most of the reviewer's comments. The reviewer would recommend the paper to the journal if the following comment can be addressed:

The layout of Section 2 does not make sense as there is only Section 2.1 while no Section 2.2, 2.3, etc. Are all discussions under Section 2 about Section 2.1? 

Author Response

(The authors gave the same response as above.)
